# The Alterations of Serum N-glycome in Response to SARS-CoV-2 Vaccination

**DOI:** 10.3390/ijms24076203

**Published:** 2023-03-25

**Authors:** Dalma Dojcsák, Zsófia Kardos, Miklós Szabó, Csaba Oláh, Zsolt Körömi, Béla Viskolcz, Csaba Váradi

**Affiliations:** 1Advanced Materials and Intelligent Technologies Higher Education and Industrial Cooperation Centre, University of Miskolc, 3515 Miskolc, Hungary; 2Borsod Academic County Hospital, 3526 Miskolc, Hungary

**Keywords:** serum glycosylation, COVID-19, vaccination, liquid chromatography, mass spectrometry

## Abstract

The severe acute respiratory syndrome coronavirus 2 (SARS-CoV-2) pandemic has caused a global concern since its outbreak in 2019, with one of the main solutions being vaccination. Altered glycosylation has been described in patients after SARS-CoV-2 infection, while the effect of vaccination on serum glycoproteins remained unexplored. In this study, total serum glycosylation was analyzed in patients after SARS-CoV-2 infection and/or mRNA vaccination in order to identify potential glycosylation-based alterations. Enzyme-linked immunosorbent assay was applied to identify post-COVID-19 and post-Vaccinated patients and rule out potential outliers. Serum samples were deglycosylated by PNGase F digestion, and the released glycans were fluorescently derivatized using procainamide labeling. Solid-phase extraction was used to purify the labeled glycans followed by the analysis of hydrophilic-interaction liquid chromatography with fluorescence and mass-spectrometric detection. Alterations of serum N-glycome in response to SARS-CoV-2 infection and mRNA vaccination were revealed by linear discriminant analysis.

## 1. Introduction

The global coronavirus disease 2019 (COVID-19) pandemic has affected millions of people worldwide, with several infection waves occurring in most countries [1]. COVID-19 is caused by the severe acute respiratory syndrome coronavirus 2 (SARS-CoV-2) targeting human cells through the angiotensin-converting enzyme 2 (ACE2) receptor [2]. The presence of ACE2 receptor on multiple cell types has resulted in the diversity of COVID-19 symptoms [3]. One of the main strategies to control the pandemic was the use of vaccination against the SARS-CoV-2 virus, reducing the duration of infection time and thus the number of COVID-19-positive cases [4]. One of the first approved immunization strategies was the use of mRNA vaccines targeting the production of antibodies against the spike protein of coronavirus and providing a defensive barrier against a potential next infection [5]. Glycosylation is a chemical modification of proteins by the covalent attachment of carbohydrate chains after translation, serving as an important signal in the quality control of protein synthesis [6]. The monosaccharide composition and the terminal sugar units have crucial impact on physicochemical properties and biological functions of the parent proteins [7]. As the synthesis of glycans is non-template driven in contrast to proteins, their composition can be altered under pathological conditions, and thus, their analysis can improve the recognition of cellular dysfunctions and serve as the signature of diseases [8,9]. Glycans are complex carbohydrates consisting of multiple monosaccharide units with no fluorophore group requiring multi-step sample preparation and high-resolution separation methods for their sensitive and reliable quantitative analysis. The most efficient separation techniques in quantitative glycomics are the ultra-performance liquid chromatography and capillary electrophoresis combined with fluorescence and/or mass-spectrometric detection allowing the implementation of large scale biomarker studies with high reproducibility [10,11]. Using these separation methods, typical alterations in the glycosylation pattern of serum N-glycome were described in numerous malignant [12] and inflammatory diseases [13]. The changes in serum N-glycosylation have also been identified in several infectious diseases including tuberculosis, HIV, influenza, ebola and viral hepatitis [14]. Recent studies suggest that serum glycosylation can be significantly altered in patients after SARS-CoV-2 infection, and the analysis of serum N-glycome might be significant in the surveillance of COVID-19 [15,16].

In this study, total serum N-glycome was analyzed in patients after SARS-CoV-2 infection and/or after mRNA vaccination in order to identify potential glycosylation-based alterations using hydrophilic-interaction liquid chromatography with fluorescence detection. Glycans from the serum samples were released by PNGase F digestion-based deglycosylation followed via fluorescent derivatization and hydrophilic solid phase extraction. Each individual patient sample was relatively quantified by fluorescence detection, and the generated peak area percentages were used to apply statistical analyses. Associated glycosylation alterations in response to SARS-CoV-2 infection and mRNA vaccination were revealed by statistical analysis.

## 2. Results and Discussion

The quality of the analyzed samples is critical in any bioanalytical studies, and thus, in order to reveal accurate glycosylation alterations, anti-SARS-CoV-2 IgG positivity was determined across the patient groups by anti-SARS-CoV-2 IgG ELISA immunoassay. The provided cut-off value by the manufacturer was 0.1, as shown in Table 1, indicating the presence or absence of reactive IgG antibodies against the SARS-CoV-2 virus. It is important to note that this step was essential to carry out this study, as we originally had a higher number of patient samples, especially in the COVID-Vaccine- group, although after the ELISA measurements, we have noticed that some of the patients were probably SARS-CoV-2 infected in the past without knowing it.

Once the 4 patient groups were defined (16 with no previous SARS-CoV-2 infection/no vaccination (COVID-Vaccine-), 16 with no previous SARS-CoV-2 infection/vaccinated (COVID-Vaccine+), 16 who underwent SARS-CoV-2 infection/no vaccination (COVID+Vaccine-), and 16 who underwent SARS-CoV-2 infection/vaccinated (COVID+Vaccine+)) based on their anti-SARS-CoV-2 IgG reactivity, total serum glycosylation was analyzed in each individual patient samples (16/group) by HILIC-UPLC. All samples were analyzed in triplicates generating 192 chromatograms with 41 individual glycan peaks, which were relatively quantified based on their area percentages.

Representative chromatograms of the pooled serum samples from COVID-Vaccine-, COVID-Vaccine+, COVID+Vaccine- and COVID+Vaccine+ are presented in Figure 1A, with the main structures highlighted. Similarly to previous reports [17,18], the glycan structures released from serum glycoproteins were mainly identified as bi-, tri- and tetra-antennary structures with various degrees of sialylation and fucosylation and, furthermore, some high-mannose structures with low abundance. The structural elucidation of high abundant structures, namely A2G2S2 (Appendix A), FA2G2S2 (Appendix A), A2G2S1 (Appendix A), FA2 (Appendix A), A3G3S3 (Appendix A), M5 (Appendix A) was performed by MS/MS analysis and subsequent annotation of their fragmentation patterns.

As shown in Figure 1B, the analyzed patient groups were well-separated based on their serum N-glycome distribution using linear discriminant analysis. The importance of the individual glycan structures is also visualized in Figure 1C, suggesting their contributions in the differentiation of the analyzed samples. The highest relative influence on the separation of the patient groups was shown by the A2G2S2 structure (Figure 1C), which was also the most abundant glycan structure with ~30% relative area on average. The level of A2G2S2 was found to be higher in both post-COVID-19 (COVID+) and post-Vaccinated (Vaccine+) patients, a trend which was also visible on most sialylated species (Figure 2A–C). A2G2S1 also had a significant impact on the separation of the patient groups with similar distribution to A2G2S2, as higher area percentages were found in the group of COVID-Vaccine+ and COVID+Vaccine- compared to COVID-Vaccine- (Figure 1C). The higher sialylation was mainly visible on non-fucosylated structures as, in the case of FA2G2S2, lower peak area percentages were detected in both post-COVID-19 and post-Vaccinated patient groups (Figure 2D), which were also typical on most fucosylated and sialylated structures (Appendix A). In contrast to the sialylated glycan species, the neutrals (no terminal sialic acid) and high-mannose structures showed a decreasing tendency across the examined groups especially in the case of FA2 and Man5, as shown in Figure 2E,F. The importance of sialylation across the examined patient groups was also shown by the fact that none of the neutral structures were found to be significantly altered (Appendix A).

Our results suggest the increase in non-fucosylated-sialylated (A3G3S3, A2G2S2, A2G2S1) glycan species in both post-COVID-19 and post-Vaccinated patients. Interestingly, the higher level of sialylation in post-COVID-19 patients was also found to be increased in response to vaccination. In case of the fucosylated and sialylated structures, there was no significant increase in response to vaccination, while higher fucosylation values were found in post-COVID-19 patients (Figure 2D). These results suggest that typical alterations of serum N-glycome after SARS-CoV-2 infection include higher sialylation and fucosylation, while in response to vaccination mainly the non-fucosylated sialylated structures are increased. It is crucial to note that the higher sialylation level in post-COVID-19 patients was also increased in response to vaccination, suggesting the importance of sialylation.

The presented results are in agreement with previous reports, where increased levels of mono-, bi-, tri- and tetra-sialylation and also higher fucosylation were found in COVID-19-positive patients compared to healthy controls [15]. Beimdiek et al. reported an elevation in the level of sialylated bi-antennary glycans supporting our results, although they have also found increased level of oligomannose structures, which was not significantly altered and rather decreased in our results. [16]. Higher sialylation in response to influenza vaccine was also documented suggesting that glycosylation alterations can improve the identification of responders [19]. The role of sialylation in the suppression of inflammatory processes has also been identified in multiple conditions [20,21]. These findings verify previous reports suggesting the importance of glycosylation monitoring not only as disease signatures but as useful markers in response to medication as well [22].

## 3. Materials and Methods

### 3.1. Chemicals

Formic acid, ammonium-hydroxide, acetic acid, acetonitrile, picoline borane, procainamide-hydrochloride and dimethyl sulfoxide were purchased from Sigma-Aldrich (St. Louis, MO, USA). Phosphate-Buffered Saline was obtained from VWR (Radnor, PA, USA). Ammonia solution was obtained from Scharlab S.L. (Barcelona, Spain). PNGase F was obtained from New England Biolabs (Ipswich, MA, USA).

### 3.2. Patient Samples

Serum samples from 64 patients (16 with no previous SARS-CoV-2 infection/no vaccination (COVID-Vaccine-), 16 with no previous SARS-CoV-2 infection/vaccinated (COVID-Vaccine+), 16 who underwent SARS-CoV-2 infection/no vaccination (COVID+Vaccine-), and 16 who underwent SARS-CoV-2 infection/vaccinated (COVID+Vaccine+)) were collected at the Borsod Academic County Hospital (Miskolc, Hungary). The blood samples were taken at least 1 month after the administration of the second dose of the Pfizer–BioNTech COVID-19 mRNA vaccine. Informed consent forms were signed by all the patients in accordance with the Declaration of Helsinki. The study was approved by the Regional Research Ethics Committee (Ethical approval number: BORS-02-2021). Clinical samples were obtained from 49 female and 15 male patients with average age of 45.2.

### 3.3. N-Glycan Release from Serum Proteins, Labelling and Clean-Up

The glycan release was performed using 9 µL of serum sample, according to the PNGase F deglycosylation protocol of New England Biolabs (Ipswich, MA, USA). The released glycans were labeled by the addition of 10 μL 0.3 M procainamide and 300 mM picoline borane in 70%/30% of dimethyl sulfoxide/acetic acid incubating for 4 h at 65 °C. The purification of labeled glycans was performed by NH_2_-functionalized MonoSpin columns (GL Sciences Inc., Tokyo, Japan) according to the manufacturer’s protocol. The purified carbohydrates were dissolved in 25%/75% water/acetonitrile and analyzed by HILIC-UPLC.

### 3.4. UPLC-FLR-MS Analysis

The prepared N-glycans were analyzed by a Waters Acquity ultra-performance liquid chromatography instrument equipped with a fluorescence detector and a Xevo-G2S qTOF mass spectrometer. The system was controlled by MassLynx 4.2 (Waters, Milford, MA, USA). Separations were performed by a Waters BEH Glycan column, 100 × 2.1 mm i.d., 1.7 μm particles, using a linear gradient of 75–55% acetonitrile (Buffer B) at 0.4 mL/min in 42 min, using 50 mM ammonium formate pH 4.4 as Buffer A. An amount of 5 μL of sample was injected using partial loop mode in all runs. The sample manager temperature was 15 °C, and the column temperature was 60 °C during each analysis. The fluorescence detection excitation and emission wavelengths were λ_ex_ = 308 nm and λ_em_ = 359 nm. During the MS analysis, 2.2 kV electrospray voltage applied to the capillary. The desolvation temperature was set to 120 °C, while the desolvation gas flow was 800 L/hr. Mass spectra were acquired using positive ionization mode over the range of 500–2000 m/z. MS/MS fragments were obtained using 45 kV collision energy during the analysis.

### 3.5. SARS-CoV-2 IgG ELISA Immunoassay

For the identification of potential outlier patients (patients who claimed to be post-COVID-19 negative although they had COVID-19-specific antibodies) SARS-CoV-2 specific IgG ELISA Immunoassay was performed using a microplate based ELISA kit (Autobio Diagnostics CO., Ltd., Zhengzhou, China). The immunoassay was performed by a chemiluminescence reaction detected as absorbance, which was proportional to the amount of SARS-CoV-2 IgG in the serum. Measurement procedure was carried out using a ClarioStar plate reader (BMG LabTech, Ortenberg, Germany) with chemiluminescence detection at 450 nm.

### 3.6. Data Analysis

Each patient sample was analyzed in triplicate, and all chromatograms were integrated by Empower 2 chromatography software (Waters, Milford, MA, USA). The mass calculation of the individual glycan structures was performed in GlycoWorkbench. Glycan nomenclature was used as it has been described by Harvey et al. [23]. The statistical analyzes were carried out in Past 4.11 software using linear discriminant analysis (LDA), EZinfo 3.0 software for the variable importance analysis (VIP) and IBM SPSS Statistics 23 to perform Kruskal–Wallis tests.

## 4. Conclusions

Total serum glycosylation was analyzed in this study in patients after SARS-CoV-2 infection and/or after mRNA vaccination in order to identify potential glycosylation alterations using HILIC-UPLC. Enzyme-linked immunosorbent assay was applied to identify post-COVID-19 and post-Vaccinated patients and rule out potential outliers. Based on our results, we have identified significantly higher sialylation levels on mono-, bi- and tri-sialylated glycan species while the fucosylated glycans showed lower levels in response to SARS-CoV-2 vaccination. Our future plan is to identify the parent proteins of the detected glycosylation alterations.

## Figures and Tables

**Figure 1 ijms-24-06203-f001:**
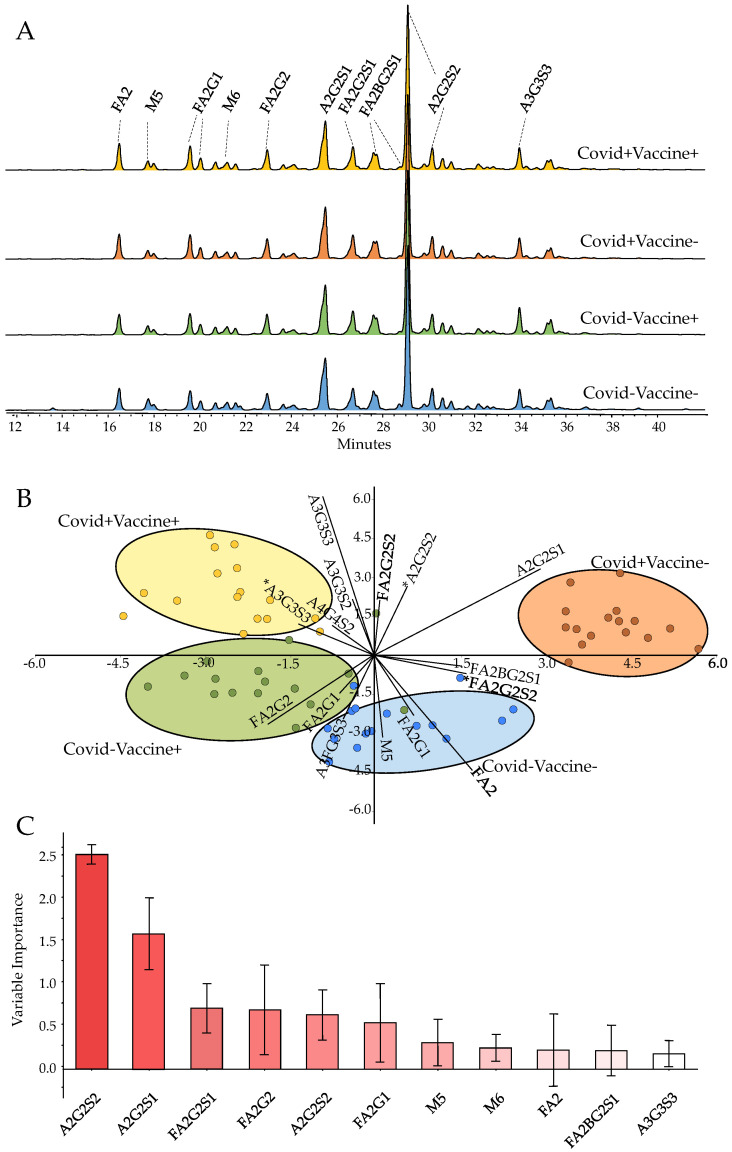
Pooled serum N-glycome in COVID-Vaccine-, COVID-Vaccine+, COVID+Vaccine- and COVID+Vaccine+ patients (**A**), linear discriminant analysis based on their N-glycome distribution (**B**) and the contributions of the glycan structures to the separation of the patient groups (**C**). (* Statistically significant alterations).

**Figure 2 ijms-24-06203-f002:**
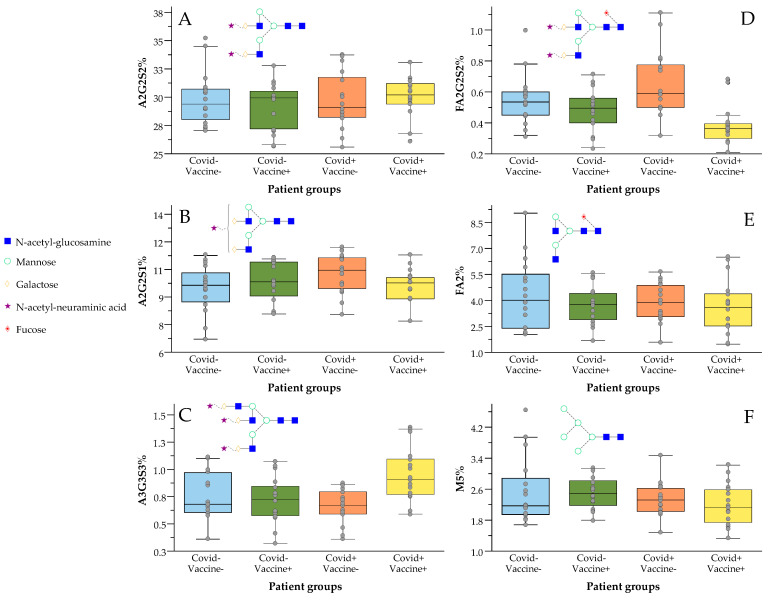
Increased sialylation and decreased fucosylation levels in response to vaccination against SARS-CoV-2 virus ((**A**) A2G2S2; (**B**) A2G2S1; (**C**) A3G3S3; (**D**) FA2G2S2; (**E**) FA2; (**F**) M5).

**Table 1 ijms-24-06203-t001:** Anti-SARS-CoV-2 specific immunoassay among the analyzed patient groups (COVID-Vaccine-, COVID-Vaccine+, COVID+Vaccine-, and COVID+Vaccine+).

Group Name	ELISA Mean Absorbance ± Std Deviation	Result (Cut-Off = 0.1)
COVID-Vaccine-	0.06 ± 0.05	non-reactive
COVID-Vaccine+	0.69 ± 0.27	reactive
COVID+Vaccine-	0.40 ± 0.28	reactive
COVID+Vaccine+	1.04 ± 0.39	reactive

## Data Availability

The generated data can be requested from the corresponding author.

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
