# Peer review of "The Alterations of Serum N-glycome in Response to SARS-CoV-2 Vaccination"

_ijms, 2023, doi:10.3390/ijms24076203_

Round 1

Reviewer 1 Report

Row 78: Explanation of the patient groups needed

Row 96: Consider rewriting the whole paragraph to improve the text flow

Row 113: Explanation of neutral structures needed

Row 147: Patient groups should be also explained in the results section

Author Response

Response to Reviewer 1 Comments

Point 1: Row 78: Explanation of the patient groups needed.

Response 1: Thank you for your recommendation. We have explained patient groups.

Point 2: Row 96: Consider rewriting the whole paragraph to improve the text flow.

Response 2: Thank you for your recommendation. The paragraph has been rewritten.

Point 3: Row 113: Explanation of neutral structures needed.

Response 3: We have explained neutral structures.

Point 4: Row 147: Patient groups should be also explained in the results section.

Response 4: We have clarified the patient groups in the results section.

Reviewer 2 Report

The short communication article deals with interesting topic of changes in N-glycosilation induced by SARS-2 CoV-2 infection or vaccination. The authors used adequate analytical methods for N-glycan analysis and study involved samples from 64 donors. The main finding of the article is that the vaccination has an impact on serum N-glycome. However, before acceptance these points needs to be addressed:

1. Table 1 - add std deviation for mean IgG concentration, add number of samples for each group

2. More thoughtful statistical analysis and more details for statistical evaluation are needed. There is stated in abstract section that the result were evaluated by multiple statistical analyses but in the main text only LDA is presented. Why didn't authors use PCA or OPLS-DA? Other methods for statistical evaluation are needed. 

3. Fig. 2 - it would be good to show also the inner points and outliners

4. Is the N-glycome somehow connected with the IgG concentration? The correlation or statistical evaluation of this relation should be discussed.

Round 2

Reviewer 2 Report

The statistical evaluation needs to be precisely discussed in manuscript. There should be results from multiple statistical analyses since no statisticaly significant results are obvious from presented graphs. The the main result comes from LDA, indicating some connection between Nglycosilation and vaccination, however no discussion and no comparison with other statistical methods is described.

Pont 3 from previous review report: Why not use modified graph from Excel? 

Author Response

The suggested modification in Figure 2 has been done based on the reviewer's comment.

Round 3

Reviewer 2 Report

The evaluation by multiple statistical analyses is mentioned even in abstract: "Fingerprints of serum N-glycome in response to SARS-CoV-2 infection and mRNA vaccination 19 were revealed by multiple statistical analyses." However only result of LDA is described in article. Addiional methods should be used and disscused in article.  

Author Response

Point 1: The evaluation by multiple statistical analyses is mentioned even in abstract: "Fingerprints of serum N-glycome in response to SARS-CoV-2 infection and mRNA vaccination 19 were revealed by multiple statistical analyses." However only result of LDA is described in article. Addiional methods should be used and disscused in article.

Response 1: Thank you for the suggestion. The abstract has been modified and the data has been analyzed by Kruskal-Wallis test (Figure2) and ROC curve analysis (Supplementary Figure 7).

Round 4

Reviewer 2 Report

Authors responded well to my comments and I recommend the publication of manuscript in IJMS.